# Detection of Static and Mobile Targets by an Autonomous Agent with Deep Q-Learning Abilities

**DOI:** 10.3390/e24081168

**Published:** 2022-08-22

**Authors:** Barouch Matzliach, Irad Ben-Gal, Evgeny Kagan

**Affiliations:** 1Department Industrial Engineering, Tel-Aviv University, 6997801 Tel Aviv, Israel; 2Laboratory for Artificial Intelligence, Machine Learning, Business and Data Analytics, Tel-Aviv University, 6997801 Tel Aviv, Israel; 3Department Industrial Engineering, Ariel University, 4076414 Ariel, Israel

**Keywords:** search and detection, probabilistic decision-making, autonomous agent, deep Q-learning, neural network

## Abstract

This paper addresses the problem of detecting multiple static and mobile targets by an autonomous mobile agent acting under uncertainty. It is assumed that the agent is able to detect targets at different distances and that the detection includes errors of the first and second types. The goal of the agent is to plan and follow a trajectory that results in the detection of the targets in a minimal time. The suggested solution implements the approach of deep Q-learning applied to maximize the cumulative information gain regarding the targets’ locations and minimize the trajectory length on the map with a predefined detection probability. The Q-learning process is based on a neural network that receives the agent location and current probability map and results in the preferred move of the agent. The presented procedure is compared with the previously developed techniques of sequential decision making, and it is demonstrated that the suggested novel algorithm strongly outperforms the existing methods.

## 1. Introduction

The detection of hidden stationary or moving targets is the first task of search procedures; this task focuses on recognizing target locations and precedes the chasing of the targets by the search agent [1,2]. Usually, the solution of the detection problem is represented by a certain distribution of the search effort over the considered domain [3,4]; for recent results and an overview of the progress in this field, see, e.g., [5,6,7,8].

In the simplest scenario of the detection of static targets by a static agent, it is assumed that the agent is equipped with a sensor that can obtain information (complete or incomplete) from all points in the domain. Using such a sensor, the agent screens the environment and accumulates information about the targets’ locations; when the resulting accumulated information becomes sufficiently exact, the agent returns a map of the domain with the marked locations of the targets.

In the case of a moving agent, the detection process acts similarly, but it is assumed that the agent is able to move over the domain to clarify the obtained information or to reach a point from which the targets can be better recognized. A decision regarding the agent’s movement is made at each step and leads the agent to follow the shortest trajectory to achieve the detection of all targets.

Finally, in the most complex scenario of moving target detection, the agent both moves within the domain to find a better observation position and tracks the targets to obtain exact information about each of their locations.

It is clear that in the first scenario, the agent has a passive role, and the problem is focused not on decision making, but on sensing and sensor fusion. However, in the case of a moving agent, the problem focuses on planning the agent’s path.

In recent decades, several approaches have been suggested for planning the agent’s motion and specifying decision-making techniques for detection tasks; for an overview of such research, see, e.g., [9,10]. Formally, such research addresses stochastic optimization methods that process offline and result in a complete agent trajectory or involve certain heuristic algorithms that allow the agent’s path to be planned in real time.

In this research, we follow the direction of heuristic algorithms for search and detection with false positive and false negative detection errors [7,11,12,13] and consider the detection of static and moving targets. In addition, we assume that the agent is equipped with an on-board controller that is powerful enough to process deep Q-learning and train neural networks on relatively large data sets. Similar to previously obtained solutions [12,13], a data set is represented by an occupancy grid [14,15], and the decision making for the probability maps follows the Bayesian approach [7,8].

The implemented deep Q-learning scheme follows general deep learning techniques [16,17] applied to search and detection processes [18] and to navigation of mobile agents [19]. However, in addition to usual functionality, the suggested method utilizes the knowledge about the targets’ locations in the form of probability map.

In the suggested algorithm, it is assumed that the agent starts with an initial probability map of the targets’ locations and makes decisions about its further movements either by maximizing the expected cumulative information gain regarding the targets’ locations or by minimizing the expected length of the agent’s trajectory up to obtaining the desired probability map. For brevity, we refer to the first approach as the Q-max algorithm and the second approach as the Shortest Path Length (SPL) algorithm.

The maximization of the expected information gain and minimization of the expected path length are performed with a conventional dynamic programming approach, while the decision regarding the next step of the agent is obtained by the deep Q-learning of the appropriate neural network. As an input, the network receives the agent location and current probability map, and the output is the preferred move of the agent. The a priori training of the network is conducted on the basis of a set of simulated realizations of the considered detection process.

Thus, the main contributions of the paper are the following. In contrast to known search algorithms with learning, the suggested algorithm allows search and detection with false positive and false negative detection errors, and, in addition to general deep learning scheme, the suggested algorithm utilizes the current agent’s knowledge about the targets’ locations. Note that both featured of the suggested algorithm can be used for solving the other problems that can be formulated in the terms of autonomous agents and probability maps.

The algorithm and the training data set were implemented in the Python programming language with the PyTorch machine learning library. The performance of the algorithm was compared with the performance of previously developed methods. It was found that the novel deep Q-learning algorithm strongly outperforms (in the sense of obtaining the shortest agent path length) the existing algorithms with sequential decision-making and no learning ability. Therefore, it allows the targets to be detected in less time than the known methods.

## 2. Problem Formulation

Let $C=\left\{c_{1},c_{2},\ldots ,c_{n}\right\}$ be a finite set of cells that represents a gridded two-dimensional domain. It is assumed that in the domain C there are ξ targets, ν=1,…,ξ, ξ≤n−1, which can stay in their locations or move over the set C, and an agent, which moves over the domain with the goal of detecting the targets.

It is assumed that the agent is equipped with an appropriate sensor such that the detection probability becomes higher as the agent moves closer to the target and as the agent observes the target location for a longer time, and the goal is to find a policy for the agent’s motion such that it detects all ξ targets in minimal time or, equivalently, it follows the shortest trajectory.

This detection problem follows the Koopman framework of search and detection problems [3] (see also [4,8]) and continues the line of previously developed heuristic algorithms [12,13].

Following the occupancy grid approach [14,15], the state $s\left(c_{i},t\right)$ of each cell $c_{i}\in C$, i=1, 2, …,n, at time t=1,2,… is considered a random variable with the values $s\left(c_{i},t\right)\in \left\{0,1\right\}$; $s\left(c_{i},t\right)=0$ means that the cell $c_{i}$ at time t is empty, and $s\left(c_{i},t\right)=1$ means that this cell $c_{i}$ at time t is occupied by a target. Since these two events are complementary, their probabilities satisfy
(1)$$Pr\left\{s\left(c_{i},t\right)=0\right\}+Pr\left\{s\left(c_{i},t\right)=1\right\}=1.$$

Detection is considered as an ability of the agent to recognize the states $s_{i}$ of the cells, i=1, 2, …,n, and it is assumed that the probability of detecting the target is governed by the Koopman exponential random search formula [3]
(2)$$Pr\left\{target\;detected\;in\;c_{i}\;|\;target\;located\;in\;c_{i}\right\}=1-\exp\left[-\kappa \left(c_{i},c_{j},\tau \right)\right],$$
where $\kappa \left(c_{i},c_{j},\tau \right)$ is the search effort applied to cell $c_{i}$ when the agent is located in cell $c_{j}$ and the observation period is τ. Usually, the search effort $\kappa \left(c_{i},c_{j},\tau \right)$ is proportional to the ratio of observation period τ to the distance $d\left(c_{i},c_{j}\right)$ between the cells $c_{i}$ and $c_{j}$, $\kappa \left(c_{i},c_{j},\tau \right)\sim \tau /d\left(c_{i},c_{j}\right)$, which represents the assumption that the shorter the distance $d\left(c_{i},c_{j}\right)$ between the agent and the observed cell and the longer the observation period τ, the higher the detection probability is.

To define the possibility of false positive and false negative detection errors, we assume that the occupied cells, the states of which at time t, t=1, 2,…, are $s\left(c,t\right)=1$, broadcast an alarm $\tilde{a}\left(c,t\right)=1$ with probability
(3)$$p_{TA}=Pr\left\{\tilde{a}\left(c,t\right)=1│s\left(c,t\right)=1\right\}.$$

The empty cells, the states of which at time t, t=1, 2,…, are $s\left(c,t\right)=0$, broadcast the alarm $\tilde{a}\left(c,t\right)=1$ with probability
(4)$$p_{FA}=Pr\left\{\tilde{a}\left(c,t\right)=1│s\left(c,t\right)=0\right\}=\alpha p_{TA},$$
where 0≤α<1. The first alarm is called a true alarm, and the second alarm is called a false alarm.

By the Koopman formula, the probability of perceiving the alarms is
(5)$$Pr\left\{alarm\;percieved\;at\;c_{j}\;|\;alarm\;sent\;from\;c_{i}\right\}=\exp\left[-d\left(c_{i},c_{j}\right)/\lambda \right],$$
where λ is the sensitivity of the sensor installed on the agent; it is assumed that all the cells are observed during the same period, so the value τ can be omitted.

Denote by $p_{i}\left(t\right)=Pr\left\{s\left(c_{i},t\right)=1\right\}$ the probability that at time t, cell $c_{i}$ is occupied by the target, that is, its state is $s\left(c_{i},t\right)=1$. The vector $P\left(t\right)=\left\{p_{1}\left(t\right),p_{2}\left(t\right),\ldots ,p_{n}\left(t\right)\right\}$ of probabilities $p_{i}\left(t\right)$, i=1,2,…,n, also called the probability map of the domain, represents the agent’s knowledge about the targets’ locations in the cells $c_{i}\in C$, i=1,2,…,n, at time t.

Then, the probability of the event ${\tilde{x}}_{j}\left(c_{i},t\right)$, i,j=1,2,…,n, that at time t the agent located in cell $c_{j}$ receives a signal from cell $c_{i}$, is defined as follows:(6)$$p\left({\tilde{x}}_{j}\left(c_{i},t\right)\right)=p_{i}\left(t-1\right)p_{TA}\exp\left[-d\left(c_{i},c_{j}\right)/\lambda \right]+\left(1-p_{i}\left(t-1\right)\right)p_{FA}\exp\left[-d\left(c_{i},c_{j}\right)/\lambda \right],$$
and the probability of the event $\bar{{\tilde{x}}_{j}\left(c_{i},t\right)}$, that this agent does not receive a signal at time t, is
(7)$$p\left(\bar{{\tilde{x}}_{j}\left(c_{i},t\right)}\right)=1-p\left({\tilde{x}}_{j}\left(c_{i},t\right)\right).$$

Note that the event ${\tilde{x}}_{j}\left(c_{i},t\right)$ represents the fact that the agent does not distinguish between true and false alarms, but it indicates that the agent receives a signal (which can be either a true or false alarm) from cell $c_{i}$. If α=1 and therefore $p_{TA}=p_{FA}$, then
(8)$$p\left({\tilde{x}}_{j}\left(c_{i},t\right)\right)=p_{TA}\exp\left[-d\left(c_{i},c_{j}\right)/\lambda \right],$$
which means that the agent’s knowledge about the targets’ locations does not depend on the probability map.

When the agent located in cell $c_{j}$ receives a signal from cell $c_{i}$, the probability that cell $c_{i}$ is occupied by the target is
(9)$$Pr\left\{s\left(c_{i},t\right)=1|{\tilde{x}}_{j}\left(c_{i},t\right)\right\}=\frac{p_{i}\left(t-1\right)Pr\left\{{\tilde{x}}_{j}\left(c_{i},t\right)|s\left(c_{i},t\right)=1\right\}}{p_{i}\left(t-1\right)Pr\left\{{\tilde{x}}_{j}\left(c_{i},t\right)|s\left(c_{i},t\right)=1\right\}+\left(1-p_{i}\left(t-1\right)\right)Pr\left\{{\tilde{x}}_{j}\left(c_{i},t\right)|s\left(c_{i},t\right)=0\right\}},$$
and the probability that $c_{i}$ is occupied by the target when the agent does not receive a signal from $c_{i}$ is
(10)$$Pr\left\{s\left(c_{i},t\right)=1|\bar{{\tilde{x}}_{j}\left(c_{i},t\right)}\right\}=\frac{p_{i}\left(t-1\right)Pr\left\{\bar{{\tilde{x}}_{j}\left(c_{i},t\right)}|s\left(c_{i},t\right)=1\right\}}{p_{i}\left(t-1\right)Pr\left\{\bar{{\tilde{x}}_{j}\left(c_{i},t\right)}|s\left(c_{i},t\right)=1\right\}+\left(1-p_{i}\left(t-1\right)\right)Pr\left\{\bar{{\tilde{x}}_{j}\left(c_{i},t\right)}|s\left(c_{i},t\right)=0\right\}},$$
where the probabilities $p_{i}\left(t-1\right)$, i=1, 2, …, n, represent the agent’s knowledge about the targets’ locations at time t−1 and it is assumed that the initial probabilities $p_{i}\left(0\right)$ at time t=0 are defined with respect to prior information; if there is any initial information about the targets’ locations, it is assumed that $p_{i}\left(0\right)=\frac{1}{2}$ for each i=1, 2, …, n.

In the framework of the Koopman approach, these probabilities are defined on the basis of Equations (6) and (7) and are represented as follows:(11)$$Pr\left\{s\left(c_{i},t\right)=1|{\tilde{x}}_{j}\left(c_{i},t\right)\right\}\;=\frac{p_{i}\left(t-1\right)p_{TA}}{p_{i}\left(t-1\right)p_{TA}+\left(1-p_{i}\left(t-1\right)\right)\alpha p_{TA}},$$
and
(12)$$Pr\left\{s\left(c_{i},t\right)=1|\bar{{\tilde{x}}_{j}\left(c_{i},t\right)}\right\}=\frac{p_{i}\left(t-1\right)\left(1-p_{TA}\exp\left[-d\left(c_{i},c_{j}\right)/\lambda \right]\right)}{p_{i}\left(t-1\right)\left(1-p_{TA}\exp\left[-d\left(c_{i},c_{j}\right)/\lambda \right]\right)+\left(1-p_{i}\left(t-1\right)\right)\left(1-\alpha p_{TA}\exp\left[-d\left(c_{i},c_{j}\right)/\lambda \right]\right)}.$$

The defined process of updating the probabilities is illustrated in Figure 1.

As illustrated by the figure, the agent receives true and false alarms through its on-board sensors, and based on this information, it updates the targets’ location probabilities with Equations (11) and (12).

In the case of static targets, the location probabilities $p_{i}\left(t\right)$, i=1, 2, …, n, depend only on the agent’s location at time t and its movements, while in the case of moving targets, these probabilities are defined both by the targets’ and by the agent’s activities. In the considered problem, it is assumed that the targets act independently on the agent’s motion, while the agent is aware of the process that governs the targets’ motion.

In general, the process of target detection is outlined as follows. At each step, the agent considers the probabilities $p_{i}\left(t\right)$, i=1, 2, …, n, for the targets’ locations and makes a decision regarding its next movement. After moving to the new location (or remaining at its current location), the agent receives signals from the available cells and updates the probabilities $p_{i}\left(t\right)$ following Equations (11) and (12). The obtained updated probabilities $p_{i}\left(t+1\right)$ are used to continue the process.

The goal is to define the motion of the agent that results in the detection of all ξ targets in minimal time. As indicated above, in detecting the targets, the agent is not required to arrive to their exact locations, but it is required to specify the locations as definitively as possible. Since finding a general definition of the optimal agent’s motion for any nontrivial scenario is computationally intractable, we are interested in a practically computable near-optimal solution.

## 3. Decision-Making Policy and Deep Q-Learning Solution

Formally, the detection problem of interest is specified as follows. Starting from the initial cell $c\left(0\right)$, at time t, the agent is located in cell $c\left(t\right)$ and makes a decision regarding its action $a\left(t\right):C\to C$ that determines to which cell $c\left(t+1\right)$ the agent should move from its current location $c\left(t\right)$.

### 3.1. The Agent’s Actions and Decision Making

We assume that the policy π:C×P→a for choosing an action does not depend on time and is specified for any t by the current agent’s location $c\left(t\right)$ and probability map $P\left(t\right)$. Then, the desired policy should produce actions such that the agent’s trajectory from the cell $c\left(0\right)$ up to the final cell $c\left(T\right)$ is as short as possible (in the sense that the termination time T is as short as possible), and that by following this trajectory, the agent detects all ξ targets. It is assumed that the number ξ of targets is not available to the agent during detection and is used to indicate the end of the detection process.

With respect to the indicated properties of the desired agent trajectory, a search for the decision-making policy can follow either the maximization of the expected cumulative information gain over the trajectory or the direct optimization of the length of the trajectory in the indicated sense of minimal detection time. The first approach is referred to as the Q-max algorithm, and the second is referred to as the SPL algorithm.

In previous research [12,13], a similar search and detection problem is solved heuristically by evaluating the decisions made at each step of the search and detection process. In the first algorithm, the agent follows the maximal Expected Information Gain (EIG) over the cells that are reachable in a single step from the agent’s current location; in the second algorithm, the agent moves one step toward the maximal expected information gain over all the cells, which is the Center Of View (COV) of the domain; and in the third algorithm, the agent moves toward the center of the distribution or the Center Of Gravity (COG) with respect to the current probability map.

In this paper, we address a more sophisticated approach that implements deep Q-learning techniques. First, we consider the information-based Q-max algorithm and then the SPL algorithm.

Let us start with the Q-max solution of the considered detection problem. Assume that at each time t the agent is located in cell $c\left(t\right)$ and action $a\left(t\right)$ is chosen from among the possible movements from cell $c\left(t\right)$, which are to step “forward”, “right-forward”, “right”, “right-backward”, “backward”, “left-backward”, “left”, or “left-forward” or “stay in the current cell”. Symbolically, we write this choice as
(13)$$a\left(t\right)\in A=\left\{↑,↗,\to ,↘,↓,↙,\leftarrow ,↖,⊙\right\}.$$

Denote by $P_{a}\left(t+1\right)$ a probability map that should represent the targets’ locations at time t+1 given that at time t, the agent chooses action $a\left(t\right)$. Then, given action $a_{t}$, the immediate expected informational reward of the agent is defined as
(14)$$R\left(a,t\right)=D_{KL}\left(P_{a}\left(t+1\right)||P\left(t\right)\right),\;$$
that is, the Kullback–Leibler distance between the map $P_{a}\left(t+1\right)$ and the current probability map $P\left(t\right)$. Given a policy π for choosing an action, the expected cumulative discounted reward obtained by an agent that starts in cell $c\left(t\right)$ with probability map $P\left(t\right)$ and chooses action $a\left(t\right)$ is
(15)$$q_{\pi }\left(c\left(t\right),P\left(t\right),a\left(t\right)\right)=E_{\pi }\left\{\sum_{\tau =0}^{\infty }\gamma^{\tau }R\left(a,t+\tau \right)\right\},$$
where as usual, the discount factor is 0<γ≤1, and the goal is to find a maximum value
(16)$$Q\left(c\left(t\right),P\left(t\right),a\left(t\right)\right)=\underset{\pi }{\max}\;q_{\pi }\left(c\left(t\right),P\left(t\right),a\left(t\right)\right)$$
of the expected reward $q_{\pi }$ over all possible policies π that can be applied after action $a\left(t\right)$ is chosen at time t.

Since the number of possible policies is infinite, the value $Q\left(c\left(t\right),P\left(t\right),a\left(t\right)\right)$ of the maximal expected cumulative discounted reward cannot be calculated exactly, and for any realistic scenario, it should be approximated. Below, we follow the deep Q-learning approach and present the Q-max algorithm, which approximates the values $Q\left(c\left(t\right),P\left(t\right),a\left(t\right)\right)$ of the reward for all possible actions (13) and therefore provides criteria for choosing the actions.

### 3.2. Dynamic Programming Scheme with Prediction and Target Neural Networks

The learning stage in the suggested Q-max algorithm is based on a neural network with dynamic programming for predicting current rewards. In the simplest configuration, which is still rather effective, the network consists of one input layer, which includes 2n neurons (recall that n is the size of the domain); one hidden layer, which also includes 2n. neurons; and an output layer, which includes #A=9 neurons with respect to the number of possible actions (13).

The inputs of the network are as follows. The first chunk of n inputs $\left(1,2,\ldots n\right)$ receives a binary vector that represents the agent location; namely, if the agent is located in cell $c_{j}$, then the jth input of the network is equal to 1 and the other n−1 inputs are equal to 0. The second chunk of n inputs $\left(n+1,n+2,\ldots 2n\right)$ receives the target location probabilities; namely, the $\left(n+i\right)$th input receives the target location probability $p_{i}$, i=1,2,…,n, as it appears in the probability map P.

The hidden layer of the network consists of 2n neurons, each of which implements a fully connected linear layer and sigmoid activation function $f\left(x\right)=1/\left(1+e^{-x}\right)$. This activation function was chosen from among several possible activation functions, such as the step function, Softplus function and SiLU function, and it was found that it provides adequate learning in all conducted simulations.

The nine neurons of the output layer correspond to the possible actions. Namely, the first output corresponds to the action $a_{1}=“↑”$, “step forward“; the second output corresponds to the action $a_{2}=“↗”$, “step right-forward”; and so on up to action $a_{9}=“⊙”$, which is “stay in the current cell”. The value of the kth output is the maximal expected cumulative discounted reward $Q\left(c_{j},P,a_{k}\right)$ obtained by the agent if it is in cell $c_{j}$, j=1,2,…,n, and given the target location probabilities $P=\left(p_{1},p_{2},\ldots ,p_{n}\right)$, it chooses action $a_{k}$, k=1,2,…,9.

The scheme of the network is shown in Figure 2.

The training stage of the network implements deep Q-learning techniques, which follow the dynamic programming approach (see, e.g., [16,17]). In general, the Bellman equation for calculating the maximal cumulative discounted reward is as follows:(17)$$Q\left(c\left(t\right),P\left(t\right),a\left(t\right)\right)=R\left(a,t\right)+\gamma \underset{a\in A}{\max}Q\left(c\left(t+1\right),P\left(t+1\right),a\right),$$
and this equation forms a basis for updating the weights of the links in the network. The data flow specified by this equation is shown in Figure 3.

Let w be a vector of the link weights of the network. In the considered case, there are $4n^{2}+18n+2n+9$ values of the weights, where $4n^{2}$ is the number of links between the input layers and the hidden layer, 18n is the number of links between the hidden layer and the output layer, 2n is the number of biases in the hidden layer and 9 is the number of biases in the output layer.

In addition, to distinguish these steps and to separate them from the real time, we enumerate the training steps by l=1,2,… below and retain the symbol t for the real-time moments of the detection process.

Denote by $Q\left(c\left(l\right),P\left(l\right),a\left(l\right);w\right)$ the maximal cumulative discounted reward calculated at step l=1,2,… by the network with weights w, and denote by $Q^{+}\left(c\left(l\right),P\left(l\right),a\left(l\right);w^{\prime }\right)$ the expected maximal cumulative discounted reward calculated using the vector $w^{\prime }$ of the updated weights following the recurrent Equation (17); that is,
(18)$$Q^{+}\left(c\left(l\right),P\left(l\right),a\left(l\right);w^{\prime }\right)=R\left(a,l\right)+\gamma \;\underset{a\in A}{\max}\;Q\left(c\left(l+1\right),P\left(l+1\right),a;w^{\prime }\right).$$
Then, the temporal difference learning error is
(19)$$\Delta_{l}\left(Q\right)=Q^{+}\left(c\left(l\right),P\left(l\right),a\left(l\right);w^{\prime }\right)-Q\left(c\left(l\right),P\left(l\right),a\left(l\right);w\right).$$

In practice, the values $Q\left(c\left(l\right),P\left(l\right),a\left(l\right);w\right)$ and $Q^{+}\left(c\left(l\right),P\left(l\right),a\left(l\right);w^{\prime }\right)$ are associated with separate neural networks; the first is called the prediction network, and the second is called the target network.

### 3.3. Model-Free and Model-Based Learning

The updating of the weights w in the prediction network is conducted by following basic backpropagation techniques; namely, the weights for the next step l+1 are updated with respect to the temporal difference learning error $\Delta_{l}\left(Q\right)$ calculated at the current step l. The weights $w^{\prime }$ in the target network are updated to the values of the weights w after an arbitrary number of iterations. In the simulations presented in subsequent sections, such updating was conducted at every fifth step.

The presented training procedure directly uses the events occurring in the environment and does not require prior knowledge about the targets’ abilities. In other words, by following this procedure, the agent detects the targets in the environment and simultaneously learns the environment and trains the neural network that supports the agent’s decision-making processes. We refer to such a scenario as model-free learning.

The actions of the presented online model-free learning procedure are illustrated in Figure 4.

Following the figure, at step l, the target location probabilities $Pr\left\{s\left(c_{i},l\right)=1|{\tilde{x}}_{j}\left(c_{i},l\right)\right\}$ and $Pr\left\{s\left(c_{i},l\right)=1|\bar{{\tilde{x}}_{j}\left(c_{i},l\right)}\right\}$, i,j=1,2,…,n, are updated by Equations (11) and (12) with respect to the events ${\tilde{x}}_{j}\left(c_{i},l\right)$ and $\bar{{\tilde{x}}_{j}\left(c_{i},l\right)}$ of receiving and not receiving a signal from cell $c_{i}$ while the agent is in cell $c_{j}$. The updated target location probabilities are used for calculating the value of the immediate reward $R\left(a,l\right)$ by Equation (14) and the value $Q^{+}\left(c\left(l\right),P\left(l\right),a\left(l\right);w^{\prime }\right)$ by Equation (18) in the target network. In parallel, the value $Q\left(c\left(l\right),P\left(l\right),a\left(l\right);w\right)$ of the prediction network is used for choosing the action and consequently for specifying the expected position of the agent in the environment. After calculating the temporal difference error $\Delta_{l}\left(Q\right)$ between the Q-values in the target and in the prediction network by Equation (19), the weights w in the prediction network are updated, and the process continues with step l+1.

Note that in all the above definitions, the cumulative reward does not depend on the previous trajectory of the agent. Hence, the process that governs the agent’s activity is a Markov process with states that include the positions of the agent and the corresponding probability maps. This property allows the use of an additional offline learning procedure based on the knowledge of the targets’ abilities.

Namely, if the abilities of the targets are known and can be represented in the form of transition probability matrices that govern the targets’ motion, the learning process can be conducted offline without checking the events occurring in the environment. In this scenario, at step l, instead of the target location probabilities $Pr\left\{s\left(c_{i},l\right)=1|{\tilde{x}}_{j}\left(c_{i},l\right)\right\}$ and $Pr\left\{s\left(c_{i},l\right)=1|\bar{{\tilde{x}}_{j}\left(c_{i},l\right)}\right\}$, i,j=1,2,…,n, the networks use the probabilities of the expected targets’ locations $Pr\left\{s\left(c_{i},l\right)=1|s\left(c_{i},l-1\right)=1\right\}$ and $Pr\left\{s\left(c_{i},l\right)=1|s\left(c_{i},l-1\right)=0\right\}$ at step l given the states of the cells at the previous step l−1.

Based on the previous definitions, these probabilities are defined as follows:(20)$$\begin{array}{l} Pr\left\{s\left(c_{i},l\right)=1|s\left(c_{i},l-1\right)=1\right\}\\ =\frac{p_{i}\left(l-1\right)p_{TA}\exp\left[-d\left(c_{i},c_{j}\right)/\lambda \right]}{p_{i}\left(l-1\right)\left(1-\alpha \right)+\alpha }+\frac{p_{i}\left(l-1\right){\left(1-\exp\left[-d\left(c_{i},c_{j}\right)/\lambda \right]\right)}^{2}}{p_{i}\left(l-1\right)\left(1-p_{TA}\exp\left[-d\left(c_{i},c_{j}\right)/\lambda \right]\right)+\left(1-p_{i}\left(l-1\right)\right)\left(1-\alpha p_{TA}\exp\left[-d\left(c_{i},c_{j}\right)/\lambda \right]\right)}, \end{array}$$
and
(21)$$\begin{array}{l} Pr\left\{s\left(c_{i},l\right)=1|s\left(c_{i},l-1\right)=0\right\}\\ =\frac{p_{i}\left(l-1\right)\alpha p_{TA}\exp\left[-d\left(c_{i},c_{j}\right)/\lambda \right]}{p_{i}\left(l-1\right)\left(1-\alpha \right)+\alpha }+\frac{p_{i}\left(l-1\right)\left(1-\exp\left[-d\left(c_{i},c_{j}\right)/\lambda \right]\right)\left(1-\alpha p_{TA}\exp\left[-d\left(c_{i},c_{j}\right)/\lambda \right]\right)}{p_{i}\left(l-1\right)\left(1-p_{TA}\exp\left[-d\left(c_{i},c_{j}\right)/\lambda \right]\right)+\left(1-p_{i}\left(l-1\right)\right)\left(1-\alpha p_{TA}\exp\left[-d\left(c_{i},c_{j}\right)/\lambda \right]\right)}.\; \end{array}$$

Since the presented procedure is based on certain knowledge about the targets’ activity, it is called the model-based learning procedure.

The actions of the model-based offline learning procedure are illustrated in Figure 5.

The model-based learning procedure differs from the model-free learning procedure only in the use of the target location probabilities and the method of updating them. In the model-free procedure, these probabilities are specified based on the events occurring in the environment. In the model-free procedure, they are calculated by following the Markov property of the system without referring to the real events in the environment.

As a result, in the model-free procedure, the learning is slower than in the model-based procedure. However, while in the first case, the agent learns during the detection process and can act without any prior information about the targets’ abilities, in the second case, it starts detection only after offline learning and requires an exact model of the targets’ activity. Thus, the choice of a particular procedure is based on the considered practical task and available information.

### 3.4. The Choice of the Actions at the Learning Stage

As indicated above, given the agent’s location $c\left(l\right)$ and the targets’ probability map $P\left(l\right)$, the neural networks used at the learning stage provide nine output Q-values that are associated with possible actions $a_{k}\in A$, k=1,2,…,9, namely,
$$Q\left(c\left(l\right),P\left(l\right),a_{1};w\right),\;Q\left(c\left(l\right),P\left(l\right),a_{2};w\right),\ldots ,Q\left(c\left(l\right),P\left(l\right),a_{9};w\right),$$
where $a_{1}=“↑”$ (“step forward”), $a_{2}=“↗”$ (“step right-forward”), and so on up to $a_{9}=“⊙”$ (“stay in the current cell”).

The choice among the actions $a_{k}\in A$ is based on the corresponding Q-values $Q\left(c\left(l\right),P\left(l\right),a_{k};w\right)$, k=1,2,…,9, and implements exploration and exploitation techniques. At the initial step l=0, when the agent has no prior learned information about the targets’ locations, action $a\left(l\right)\in A$ is chosen randomly. Then, after processing the step prescribed by action $a\left(l\right)$, the next action $a\left(l+1\right)$ is chosen either on the basis of the target location probabilities learned by the neural networks or randomly from among the actions available at this step. The ratio of random choices decreases with the number of steps, and after finalizing the learning processes in the neural networks, the actions are chosen with respect to the Q-values only.

Formally, this process can be defined using different policies, for example, with a decaying ϵ-greedy policy that uses the probability ϵ, which decreases with the increase in the number of steps from its maximal value ϵ=1 to the minimal value ϵ=0. The agent chooses an action randomly with probability ϵ and according to the greedy rule $\arg\underset{a\in A}{\max}Q\left(c\left(l\right),P\left(l\right),a;w\right)$ with probability 1−ϵ. In this policy, the choice of the action is governed by the probability ϵ and does not depend on the Q-values of the actions.

A more sophisticated policy of intermittence between random and greedy choice is the SoftMax policy. In this policy, the probability $p\left(a_{k}|Q;\eta \right)$ of choosing action $a_{k}$ is defined with respect to both the parameter $\eta \in \left.\left[0,+\infty \right.\right)$ and the Q-values of the actions:(22)$$p\left(a_{k}|Q;\eta \right)=\frac{\exp\left[Q\left(c\left(l\right),P\left(l\right),a_{k};w\right)/\eta \right]}{\sum_{j=1}^{9}\exp\left[Q\left(c\left(l\right),P\left(l\right),a_{j};w\right)/\eta \right]}.$$
Therefore, if η→0, then $p\left(a_{k}|Q;\eta \right)\to 1$ for $a_{k}=\arg\underset{a\in A}{\max}Q\left(c\left(l\right),P\left(l\right),a;w\right)$ and $p\left(a_{k}|Q;\eta \right)\to 0$ for all other actions, and if η→∞, then $p\left(a_{k}|Q;\eta \right)\to \frac{1}{9}$, which corresponds to a randomly chosen action. The intermediate values 0<η<∞ correspond to the probabilities $p\left(a|Q;\eta \right)\in \left(0,1\right)$ and govern the randomness of the action choice. In other words, the value of the parameter η decreases with the increasing number of steps l from its maximal value to zero; thus, for the unlearned networks, the agent chooses actions randomly and then follows the information about the targets’ locations learned by the networks. The first stages with randomly chosen actions are usually interpreted as exploration stages, and the later stages based on the learned information are considered exploitation stages. In the simulations, we considered both policies and finally implemented the SoftMax policy since it provides more correct choices, especially in cases with relatively high Q-values associated with different actions.

### 3.5. Outline of the Q-Max Algorithm

Recall that according to the formulation of the detection problem, the agent acts in the finite two-dimensional domain $C=\left\{c_{1},c_{2},\ldots ,c_{n}\right\}$ and moves over this domain with the aim of detecting ξ≤n−1 hidden targets. At time t=0,1,2,… in cell $c\left(t\right)\in C$, the agent observes the domain (or, more precisely, screens the domain with the available sensors) and creates the probability map $P\left(t\right)=\left\{p_{1}\left(t\right),p_{2}\left(t\right),\ldots ,p_{n}\left(t\right)\right\}$, where $p_{i}\left(t\right)$ is the probability that at time t, cell $c_{i}$ is occupied by a target, i=1,2,…,n. Based on the probability map $P\left(t\right)$, the agent chooses an action $a\left(t\right)\in A$. By processing the chosen action $a\left(t\right)$, the agent moves to the next cell $c\left(t+1\right)$, and this process continues until the targets’ locations are detected. The agent’s goal is to find a policy of choosing the action that provides the fastest possible detection of all the targets with a predefined accuracy.

In contrast to the recently suggested algorithms [12,13], which directly implement one-step decision making, the presented novel algorithm includes learning processes and can be used both with model-free learning for direct online detection and with model-based learning for offline policy planning and further online applications of this policy. Since both learning processes follow the same steps (with the only difference being in the source of information regarding the targets’ location probabilities), below, we outline the Q-max algorithm with model-based learning.

The Q-max algorithm with model-based learning includes three stages: in the first stage, the algorithm generates the training data set, which includes reasonable probability maps with possible agent locations; in the second stage, it trains the prediction neural network using the generated data set; and in the third stage, the algorithm solves the detection problem by following the decisions made by the trained prediction neural network. Algorithm 1 outlines the first stage that is generating of the training data set.
**Algorithm 1.** Generating the training data set**Input:** domain $C=\left\{c_{1},c_{2},\ldots ,c_{n}\right\}$,  set $A=\left\{↑,↗,\to ,↘,↓,↙,\leftarrow ,↖,⊙\right\}$ of possible actions,  probability $p_{TA}$ of true alarms (Equation (3)),  rate α of false alarms and their probability $p_{FA}=\alpha p_{TA}$ (Equation (4)),  sensor sensitivity λ,  range $\left[\xi_{1},\xi_{2}\right]$ of possible numbers $0<\xi_{1}<\xi_{2}\leq n-1$ of targets,  length $L\in \left(0,\infty \right)$ of the agent’s trajectory,  number $N\in \left(0,\infty \right)$ of agent trajectories,  initial probability map $P\left(0\right)$ on the domain C.**Output:** data set that is an L×N table of pairs $\left(c,P\right)$ of agent positions c and corresponding probability maps P.1. Create the L×N data table.2. For each agent trajectory j=1,…,N do:3. Choose a number $\xi \in \left[\xi_{1},\xi_{2}\right]$ of targets according to a uniform distribution on the interval $\left[\xi_{1},\xi_{2}\right]$.4. Choose the target locations $c_{1},\;c_{2},\ldots ,c_{\xi }\in C$ randomly according to the uniform distribution on the domain C.5. Choose the initial agent position $c\left(0\right)\in C$ randomly according to the uniform distribution on the domain C.6. For l=0,…, L−1 do:7. Save the pair $\langle c\left(l\right),P\left(l\right)\rangle$ as the jth element of the data table.8. Choose an action $a\left(l\right)\in A$ randomly according to the uniform distribution on the set A.9. Apply the chosen action and set the next position $c\left(l+1\right)=a\left(c\left(l\right)\right)$ of the agent.10. Calculate the next probability map $P\left(l+1\right)$ with Equations (20) and (21).11. End for12. End for13. Return the data table.

The data training data set includes N random trajectories of length L. Each element of the data set is a pair of an agent position and a probability map.

The reason for generating the data instead of drawing it randomly is that the training data set is used at the learning stage of the prediction network, so it should represent the data in as realistic a form as possible. Since in the generated data set, the agent’s positions are taken from the connected trajectory and the corresponding probability maps are calculated with respect to these positions, possible actions, sensing abilities and environmental conditions, it can be considered a good imitation of real data.

The generated agent positions and corresponding probability maps are used as an input of the prediction neural network in the training stage. The goal of the training is specified by the objective probability map $P^{*}=\left\{p_{1}^{*},p_{2}^{*},\ldots ,p_{n}^{*}\right\}$, which defines the target location probabilities that provide sufficient information for the immediate detection of all the targets. In the best case, we have probabilities $p_{i}^{*}\in \left\{0,1\right\}$, and in practical scenarios, it is assumed that either $p_{i}^{*}\in \left[0,\varepsilon \right]$ or $p_{i}^{*}\in \left[1-\varepsilon ,1\right]$ for certain 0<ε≪1, i=1,2,…,n.

The training stage of the Q-max algorithm is implemented in the form of Algorithm 2, which is outlined below (the scheme of the learning procedure is shown in Figure 5).
**Algorithm 2.** Training the prediction neural network**Network structure:**  input layer: 2n neurons (n agent positions and n target location probabilities, both relative to the size n of the domain),  hidden layer: 2n neurons,  output layer: 9 neurons (in accordance with the number of possible actions).**Activation function:**  sigmoid function $f\left(x\right)=1/\left(1+e^{-x}\right)$.**Loss function:**  mean square error (MSE) function.**Input:** domain $C=\left\{c_{1},c_{2},\ldots ,c_{n}\right\}$,  set $A=\left\{↑,↗,\to ,↘,↓,↙,\leftarrow ,↖,⊙\right\}$ of possible actions,  probability $p_{TA}$ of true alarms (Equation (3)),  rate α of false alarms and their probability $p_{FA}=\alpha p_{TA}$ (Equation (4)),  sensor sensitivity λ,  discount factor γ,  objective probability map $P^{*}$ (obtained by using the value ε),  number r of iterations for updating the weights,  initial value η (Equation (22)) and its discount factor δ,  learning rate ρ (with respect to the type of optimizer),  number M of epochs,  initial weights w of the prediction network and initial weights $w^{\prime }=w$ of the target network,  training data set (that is, the L×N table of $\left(c,P\right)$ pairs created by Procedure 1).**Output:** The trained prediction network.1. Create the prediction network.2. Create the target network as a copy of the prediction network.3. For each epoch j=1,…,M do:4. For each pair $\left(c,P\right)$ from the training data set, do:5. For each action a∈A do:6. Calculate the value $Q\left(c,P,a;w\right)$ with the prediction network.7. Calculate the probability $p\left(a|Q;\eta \right)$ (Equation (22)).8. End for.9. Choose an action according to the probabilities $p\left(a|Q;\eta \right)$.10. Apply the chosen action and set the next position $c^{\prime }=a\left(c\right)$ of the agent.11. Calculate the next probability map $P^{\prime }$ with Equations (20) and (21).12. If $P=P^{*}$ or $c^{\prime }\notin C$, then13. Set the immediate reward $R\left(a\right)=0$.14. Else15. Calculate the immediate reward $R\left(a\right)$ with respect to P and $P^{\prime }$ (Equation (14)).16. End if.17. For each action a∈A do:18. If $P=P^{*}$ then19. Set $\;Q\left(c^{\prime },P^{\prime },a;w^{\prime }\right)=0$.20. Else21. Calculate the value $Q\left(c^{\prime },P^{\prime },a;w^{\prime }\right)$ with the target network.22. End if.23. End for.24. Calculate the target value $Q^{+}=R\left(a\right)+\gamma \underset{a\in A}{\max}Q\left(c^{\prime },P^{\prime },a;w^{\prime }\right)$ (Equation (17)).25. Calculate the temporal difference learning error as $\Delta_{l}\left(Q\right)=Q^{+}-Q\left(c,P,a;w\right)$ for the chosen action a (Equation (19)) and set $\Delta_{l}\left(Q\right)=0$ for all other actions.26. Update the weights w in the prediction network by backpropagation with respect to the error $\Delta_{l}\left(Q\right)$.27. Every r iterations, set the weights of the target network as $w^{\prime }=w$.28. End for.

The validation of the network was conducted on a validation data set that includes the pairs $\left(c,P\right)$, which are similar to the pairs appearing in the training data set but were not used in the training procedure; the size of the validation data set is approximately ten percent of the size of the training data set.

After training, the Q-max algorithm can be applied to simulated data or in a real search over a domain. It is clear that the structure of the algorithm mimics the search conducted by rescue and military services: first, the algorithm learns the environment (by itself or at least by using the model) and then continues with the search in the real environment, where the probability map is updated with respect to the received alarms and acquired events (Equations (11) and (12)) and decision-making is conducted using the prediction network.

### 3.6. The SPL Algorithm

Now let us consider the SPL algorithm. Formally, it follows the same ideas and implements the same approach as the Q-max algorithm, but it differs in the definition of the goal function. In the SPL algorithm, the goal function directly represents the aim of the agent to detect all the targets in a minimal number of steps or to take a minimal number of actions before reaching the termination condition.

In parallel to the reward $R\left(a,t\right)$ defined by Equation (14) for action a∈A conducted at time t, we define the penalty or the price paid by the agent for action a∈A at time t. In the case of the shortest path length, the payoff represents the steps of the agent; that is,
(23)$$O\left(a,t\right)=1$$
for each time t=1,2,… until termination of the search. Note again that even if the agent chooses to stay at its current position, the payoff is calculated as 1.

Then, given a policy π for choosing an action, the expected cumulative payoff of an agent that starts in cell $c\left(t\right)$ with probability map $P\left(t\right)$ and chooses action $a\left(t\right)$ is
(24)$$pl_{\pi }\left(c\left(t\right),P\left(t\right),a\left(t\right)\right)=E_{\pi }\left\{\sum_{\tau =0}^{\infty }O\left(a,t+\tau \right)\right\},$$
and the goal is to find the minimum value
(25)$$SPL\left(c\left(t\right),P\left(t\right),a\left(t\right)\right)=\underset{\pi }{\min}\;pl_{\pi }\left(c\left(t\right),P\left(t\right),a\left(t\right)\right)$$
of the expected payoff $pl_{\pi }$ over all possible policies π that can be applied after action $a\left(t\right)$ is chosen at time t.

Then, the Bellman equation for calculating the defined minimal expected path length is
(26)$$SPL\left(c\left(t\right),P\left(t\right),a\left(t\right)\right)=O\left(a,t\right)+\underset{a\in A}{\min}\;SPL\left(c\left(t+1\right),P\left(t+1\right),a\right),$$
and the equations that define the training and functionality of the neural networks follow this equation and have the same form as in the Q-max algorithm (with the obvious substitution of maximization by minimization and the use of γ=1).

## 4. Simulation Results

The suggested algorithm was implemented and tested in several scenarios and its functionality was compared with the functionality of previously developed heuristic algorithms and, in certain simple setups, with the algorithm that provides optimal solution. Numerical simulations include training of neural network, simulation of the detection process by Q-max and SPL algorithms and their comparisons with heuristic and optimal solutions.

Numerical simulations were implemented using basic tools of the Python programming language with the PyTorch machine learning library, and the trials were run on a PC Intel^®^ Core™ i7-10700 CPU with 16 GB RAM. In the simulations, the detection was conducted over a gridded square domain of size $n=n_{x}\times n_{y}$ cells, and it was assumed that the agent and each target could occupy only one cell. Given this equipment, we measured the run time of the simulations for different datasets, which demonstrated that the suggested algorithms are implementable on usual computers and do not require specific apparats for their functionality.

### 4.1. Network Training in the Q-Max Algorithm

First, let us consider the simulation of the network training. The purpose of these simulations is to verify the training method and demonstrate a decrease in the temporal difference learning error $\Delta \left(Q\right)$ with an increasing number of learning epochs. Since the network training is the same for both the Q-max and SPL algorithms, we consider the training for the Q-max algorithm.

The training data set was generated using the parameters n=10×10=100, $p_{TA}=1$, α=0.5, λ=15, $\xi_{1}=1$, $\xi_{2}=10$, L=50, N=200 and $p_{i}\left(0\right)=0.05$, i=1,2,…,n. The size of the training data set was 10000.

The input parameters in the simulation used the same values of n=10×10=100, $p_{TA}=1$, α=0.5, and λ=15, and we also specified γ=0.9 and $P^{*}$ with ε=0.05, r=5, η=100000, δ=0.99 and ρ=0.001. The number of epochs in the simulation was M=30.

The initial weights w were generated by the corresponding procedures of the PyTorch library. The optimizer used in the simulation was the ADAM optimizer from the PyTorch library.

The average time required for training the prediction neural network was approximately 10 min (on the PC described above), which is a practically reasonable time for an offline procedure. Note that after offline training, online decision-making is conducted directly by immediate choice without additional calculations.

The results of the simulations are shown in Figure 6. The presented graph was obtained by averaging the temporal difference learning errors over 10,000 pairs in the data set.

The temporal difference learning error decreases both in the training stage and in the validation stage of the learning process, and the smoothed graphs for both stages are exponential graphs with similar rates of decrease. This validates the effectiveness of the learning process and shows that progress in the network training leads to better processing of previously unknown data from the validation data set.

### 4.2. Detection by the Q-Max and SPL Algorithms

In the next simulations, we considered the detection process with the proposed Q-max and SPL algorithms and compared both algorithms with random detection, which provides the lower bound of the cumulative reward (for the Q-max algorithm) and payoff (for the SPL algorithm).

Both algorithms used the same neural network as above, and the random detection process was initialized with the same parameters as above. However, for better comparison, in the simulations of both algorithms and of random detection, we used the same number of targets ξ=2, which were located at the points $\left(5,0\right)$ and $\left(0,9\right)$, and the initial position of the agent was $c\left(0\right)=\left(9,4\right)$. By choosing these positions of the targets and the agent, it is easy to demonstrate (a) the difference between the search processes (in which the agent first moves to the closer target and then to the distant target) and the detection process (in which the agent moves to the point that provides the best observation of both targets) and (b) the motion of the agent over the domain to maximize the immediate reward or minimize the immediate payoff.

The results of the simulations are shown in Figure 7. Figure 7a shows the discounted cumulative reward for the Q-max algorithm in comparison with that of the random detection process, and Figure 7b shows similar graphs for the SPL algorithm and the random detection process.

The detection by the proposed algorithms is much better than the detection by the random procedure. Namely, the Q-max algorithm results in 20.5 units of discounted cumulative reward, while the random procedure achieves only 13.4 units of discounted reward in the same number of steps. In other words, the Q-max algorithm is nearly 1.5 times more effective than the random procedure. Similarly, while the random procedure requires 40 steps to detect the targets, the SPL algorithm requires only 20 steps, which means that the SPL algorithm is 50% better than the random procedure.

From these comparisons, it follows that the suggested algorithms outperform the random procedure in terms of both the informational reward and the agent’s path length. However, as follows from the next simulations, the numbers of agent actions up to termination in the Q-max and SPL algorithms are statistically equal, allowing either algorithm to be applied with respect to the considered practical task.

### 4.3. Comparison between the Q-Max and SPL Algorithms and the Eig, Cov and Cog Algorithms

The third set of simulations included comparisons of the suggested Q-max and SPL algorithms with the previously developed heuristic methods [12,13], which implement one-step optimization.

The simplest algorithm is based on the expected information gain, which is an immediate expected information reward
(27)$$EIG\left(a,t\right)=R\left(a,t\right)=D_{KL}\left(P_{a}\left(t+1\right)||P\left(t\right)\right).$$
as above, here, $P_{a}\left(t+1\right)$ stands for the probability map that is expected to represent the targets’ locations at time t+1 given that at time t, the agent chooses action $a\left(t\right)\in A$ and $P\left(t\right)$ is the current probability map. Then, the next action is chosen as
(28)$$a\left(t+1\right)=\arg\underset{a\in A}{\max}EIG\left(a,t\right).$$

A more sophisticated algorithm addresses the center of view, which is defined as the cell in which the agent can obtain the maximal expected information gain
(29)$$COV\left(t\right)=\arg\underset{c\in C}{\max}D_{KL}\left(P_{c}\left(t+1\right)||P\left(t\right)\right),$$
where $P_{c}\left(t+1\right)$ is a probability map that is expected to represent the targets’ locations at time t+1 when the agent is located in cell c. Then, the next action is chosen as
(30)$$a\left(t+1\right)=\arg\underset{a\in A}{\min}\;d\left(COV\left(t\right),a\left(c\left(t\right)\right)\right),$$
where $d\left(COV\left(t\right),a\left(c\left(t\right)\right)\right)$ is the distance between the center of view $COV\left(t\right)$ and cell $a\left(c\left(t\right)\right)$, to which the agent moves from its current location $c\left(t\right)$ when it executes action a. Note that in contrast to the next location $c\left(t+1\right)$, which is one of the neighboring cells of the current agent location $c\left(t\right)$, the center of view $COV\left(t\right)$ is a cell that is chosen from among all n cells of the domain.

Finally, in the third algorithm, the next action is chosen as
(31)$$a\left(t+1\right)=\arg\underset{a\in A}{\min}\;d\left(COG\left(t\right),a\left(c\left(t\right)\right)\right),$$
where $COG\left(t\right)$ stands for the “center of gravity”, which is the first moment of the probability map $P\left(t\right)$, and the remaining terms have the same meanings as above.

The Q-max and SPL algorithms used the same neural network as above and were initialized with the same parameters. As above, for all the algorithms, the agent started in the initial position $c\left(0\right)=\left(9,4\right)$ and moved over the domain with the aim of detecting ξ=2 targets.

The first simulations addressed the detection of static targets, which, as above, were located at points $\left(5,0\right)$ and $\left(0,9\right)$.

The results of the detection by different algorithms are summarized in Table 1. The results represent the averages over 30 trials for each algorithm.

The table shows that the proposed Q-max and SPL algorithms outperform previously developed methods in terms of both the number of agent actions and the value of the discounted cumulative information gain.

The results of the simulations over time are shown in Figure 8. Figure 8a shows the discounted cumulative reward for the Q-max algorithm in comparison with the COV algorithm (the best heuristic algorithm) and the random detection process, and Figure 8b shows similar graphs for the SPL algorithm compared to the COV algorithm and the random detection process.

The detection by the suggested algorithms is better than the detection by the COV algorithm. Namely, the Q-max algorithm results in 20.5 units of discounted cumulative reward, while the COV algorithm obtains 17.5 units of discounted reward in the same number of steps. In other words, the Q-max algorithm is nearly 1.15 times more effective than the COV algorithm. Similarly, while the COV algorithm requires 25 steps to detect the targets, the SPL algorithm requires only 20 steps, which means that the SPL algorithm is 25% better than the COV algorithm.

The second simulations addressed the detection of moving targets, which started in the initial positions $\left(5,0\right)$. and $\left(0,9\right)$. Regarding the targets’ motion, it is assumed that both of them, at each time t=1,2,…, can apply one of the possible actions from the set $A=\left\{↑,↗,\to ,↘,↓,↙,\leftarrow ,↖,⊙\right\}$ so that the probability of the action ⊙ is $Pr\left\{a\left(t\right)=⊙\right\}=0.9$ and the probability of each other action a∈A\⊙ is $\left(1-0.9\right)/8=0.0125$.

The results of detection by different algorithms (averaged over 30 trials for each algorithm) are summarized in Table 2.

In the detection of moving targets, the suggested Q-max and SPL algorithms also outperform previously developed methods in terms of both the number of agent actions and the value of the discounted cumulative information gain.

Note that the simulation was conducted for targets with a clear motion pattern, where the probabilities of the targets’ actions represent slow random motion of the targets near their initial locations. Another possible reasonable motion pattern is motion with a strong drift in a certain direction, which results in a similar ratio between the numbers of actions and the discounted cumulative information gains to that presented in Table 2.

In contrast, if the random motion of the targets is a random walk with equal probabilities $Pr\left\{a\left(t\right)=a\right\}=1/9$ for all actions a∈A, then the training becomes meaningless since both with and without training, the agent needs to detect randomly moving targets.

The other results obtained for the Q-max/SPL algorithms also indicated better performance by these algorithms compared with that of the heuristic algorithms. The algorithms were compared with the best heuristic COV algorithm. The results of the trials for different values of the false alarm rate α and of the sensor sensitivity λ are summarized in Table 3.

For all values of the false alarm rate and the sensor sensitivity, the Q-max and SPL algorithms strongly outperform the best heuristic COV algorithm.

To emphasize the difference in the detection time between the suggested SPL and Q-max algorithms and the heuristic COV algorithm, the data shown in the table are depicted in Figure 9.

As expected, the Q-max and SPL learning algorithms demonstrate better performance than the heuristic COV algorithms without learning, and the difference between the algorithms increases as the false alarm rate α increases and the sensor sensitivity λ decreases. For example, if λ=15 and α=0.25, then the improvement in the number of actions is 10%, while if λ=5 and α=0.75, then the improvement is significantly stronger at 75%.

In other words, computationally inexpensive heuristic algorithms provide effective results in searches with accurate sensors and a low rate of false alarms. However, in searches with less precise sensors or with a high rate of false-positive errors, the heuristic algorithms are less effective, and the Q-max and SPL learning algorithms should be applied.

### 4.4. Comparison between the SPL Algorithm and an Algorithm Providing the Optimal Solution

The suggested approach was compared with the known dynamic programming techniques implemented in search algorithms for moving targets [11,20]. Since the known algorithms directly address the optimal trajectory of the agent and result in an optimal path, in the simulation, we considered the SPL algorithm, which uses the same criteria as the known algorithms.

The comparisons were conducted as follows. The algorithms were trialed over the same domain with a definite number n of cells, and the goal was to reach the maximal probability $P^{*}=0.95$ of detecting the target. When this probability was reached, the trial was terminated, and the number of agent actions was recorded.

Since the known algorithms [11,20] implement dynamic programming optimization over possible agent trajectories, their computational complexity is high, and for the considered task, it is $O\left(n\cdot 9^{t}\right)$, where n is the number of cells and t is the number of actions.

Therefore, to finish the simulations in reasonable time (120 min for each trial), the algorithms were trialed on a very small case with n=10×10=100 cells. Note that in the original simulations, these algorithms were trialed on smaller cases. If the desired probability $P^{*}=0.95$ of detecting the targets was not reached in 120 min, the algorithms were terminated.

In all trials, the known dynamic programming algorithms planned t=7 agent actions in 120 min, while the suggested SPL algorithm, in the same time of 120 min, planned significantly more actions and reached at least the desired probability $P^{*}=0.95$ of detecting the targets. The results of the comparison between the SPL algorithm and the known dynamic programming algorithms that provide optimal solutions are summarized in Table 4.

Until termination at 120 min, the SPL algorithm plans more agent actions and results in higher detection probabilities than the DP algorithm for both values of sensor sensitivity λ and for all values of the false alarm rate α. For example, the dependence of the detection probabilities on the run time for sensor sensitivity λ=15 and false alarm rate α=0.25 is depicted in Figure 10.

For the first 7 actions, the detection probabilities of both algorithms increase similarly. Then, the DP algorithm does not plan additional actions in 120 min, while the SPL algorithm results in more planned actions, and the detection probabilities for these actions continue increasing until termination after 13 planned actions.

Finally, the dependence of the detection probabilities on the false alarm rate α at termination after 120 min is depicted in Figure 11.

For a low false alarm rate α, the SPL algorithm results in the same detection probabilities as the optimal DP algorithms, but for a higher false alarm rate α, the detection probabilities obtained by the DP algorithms significantly decrease (to 0.68 and 0.43 for λ=15 and λ=10, respectively), while the probability obtained by the SPL algorithm is 0.95 for any false alarm rate and both sensor sensitivities.

### 4.5. Run Times and Mean Squared Error for Different Sizes of Data Sets

Finally, we considered the dependence of the run time and mean squared error on the size of the data set. The results of these simulations are summarized in Table 5.

While the size of the domain and the number of links in the network exponentially increase, the mean squared error increases very slowly and remains small. In addition, it is seen that with an exponentially increasing domain size, the run time increases linearly, and the computations require a reasonable time even on the previously described PC. However, for realistic engineering and industrial problems with larger domains, it is reasonable to use computation systems with greater GPU power.

## 5. Discussion

This paper presents a novel algorithm for the navigation of mobile agents detecting static and moving hidden targets in the presence of false-positive and false-negative errors. The suggested algorithm continues in the direction of previously developed procedures [12,13] for seeking and detecting hidden targets. However, in contrast to these procedures, which follow an immediate one-step decision making process, the proposed method implements the deep Q-learning approach and neural network techniques.

The suggested algorithm is implemented in two versions: a procedure that maximizes the cumulative discounted expected information gain over the domain (Q-max algorithm) and a procedure that minimizes the expected path length of the agent in detecting all the targets (SPL algorithm). Formally, the first procedure is an extension of previously developed techniques based on the expected information gain calculated over the local neighborhood of the agent, while the second is a direct application of Q-learning techniques to the required value of the agent’s path length.

The simulations show that after offline training of the neural network using the generated data set, the algorithm provides solutions that outperform the results obtained by the previously developed procedures, both in terms of the cumulative information gain and in terms of the agent’s path length. Moreover, the expected number of actions obtained by the Q-max algorithm by maximizing the cumulative discounted expected information gain is statistically equal to the number of actions obtained by the SPL algorithm by minimizing the expected path length. This equivalence follows directly from the nature of the problem: in terms of information, the detection of the targets means accumulating as much information as possible about the targets’ locations, and in terms of the path length, the detection of the targets means making as few movements as possible in order to specify the exact target locations.

## 6. Conclusions

This paper considered the detection problem for multiple static and moving targets hidden in a domain, directly extending the classical Koopman search problem. Following previously developed methods, we addressed detection with both false-positive and false-negative detection errors.

In the exploration stage, the suggested algorithm implements the deep Q-learning approach and applies neural network techniques for learning the probabilities of the targets’ locations and their motion patterns; then, in the exploitation stage, it chooses actions based on the decisions made by the trained neural network.

The research suggested two possible procedures. In the first, called the model-free procedure, the agent detects the targets in the environment and simultaneously, online, learns the environment and trains a neural network that supports the agent’s decision-making processes. In the second procedure, called the model-based procedure, the agent begins detection only after offline learning and requires an exact model of the targets’ activity.

The results obtained by maximizing the discounted cumulative expected information gain and by minimizing the expected length of the agent’s path demonstrate that the suggested algorithm outperforms previously developed information-based procedures and provides a nearly optimal solution even in cases in which the existing techniques require an unreasonable computation time.

The proposed algorithms were implemented in the Python programming language and can be used both for further development of the methods of probabilistic search and detection and for practical applications in the appropriate fields.

## Figures and Tables

**Figure 1 entropy-24-01168-f001:**
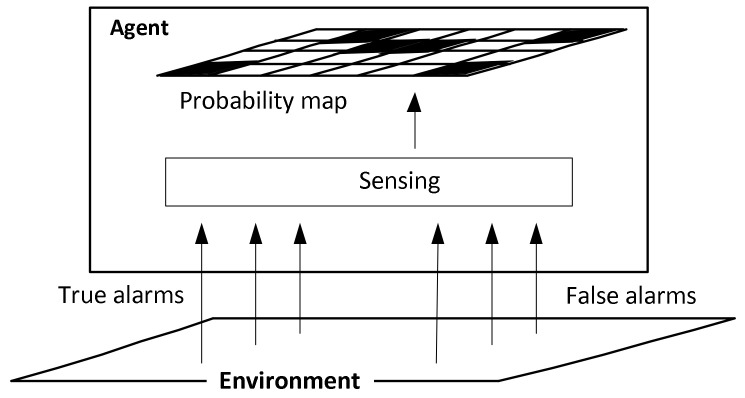
Receiving information and updating the probability map.

**Figure 2 entropy-24-01168-f002:**
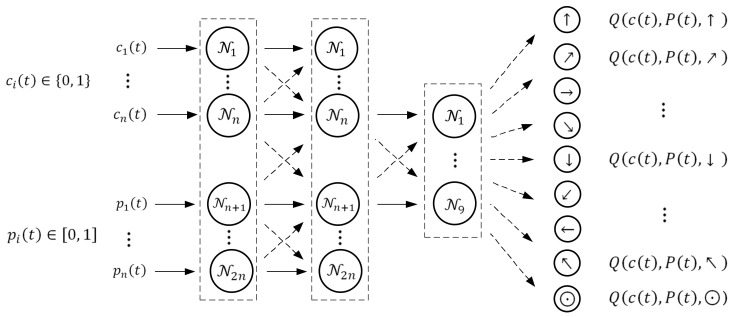
The neural network scheme used in the learning stage of the Q-max algorithm.

**Figure 3 entropy-24-01168-f003:**
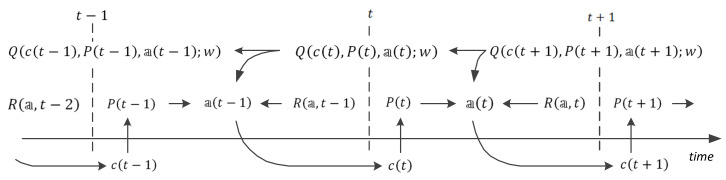
Scheme of the data flow in the training stage of the network.

**Figure 4 entropy-24-01168-f004:**
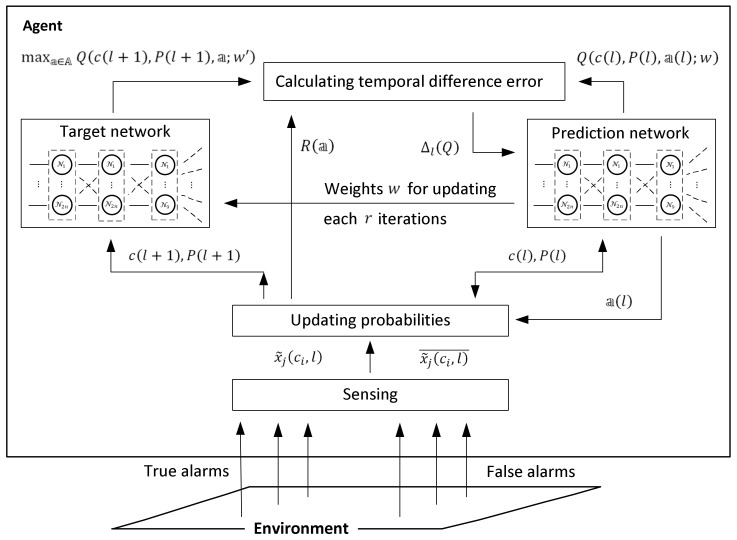
The actions of the online model-free learning procedure of the Q-max algorithm.

**Figure 5 entropy-24-01168-f005:**
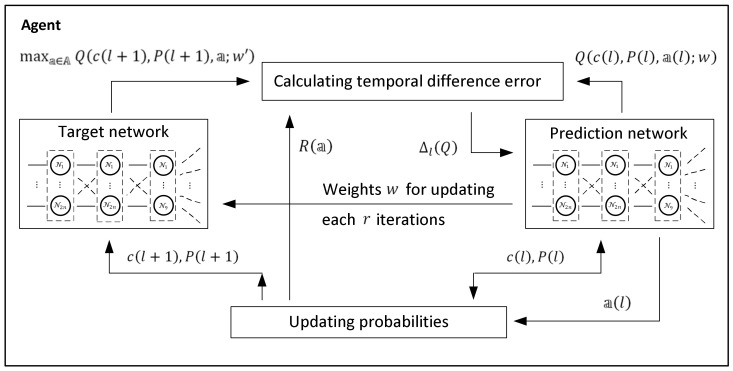
The actions of the offline model-based learning procedure of the Q-max algorithm.

**Figure 6 entropy-24-01168-f006:**
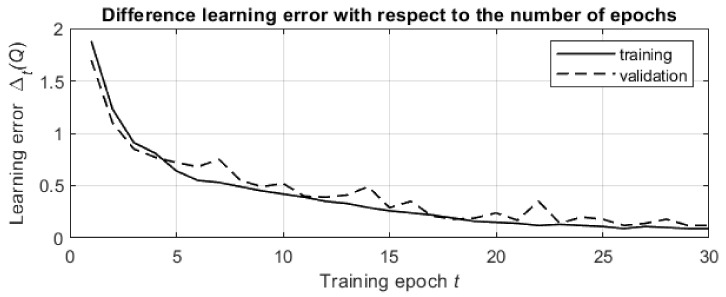
The change in the temporal difference learning error with respect to the number of training epochs. The solid line is associated with the training stage, and the dashed line is associated with the validation stage.

**Figure 7 entropy-24-01168-f007:**
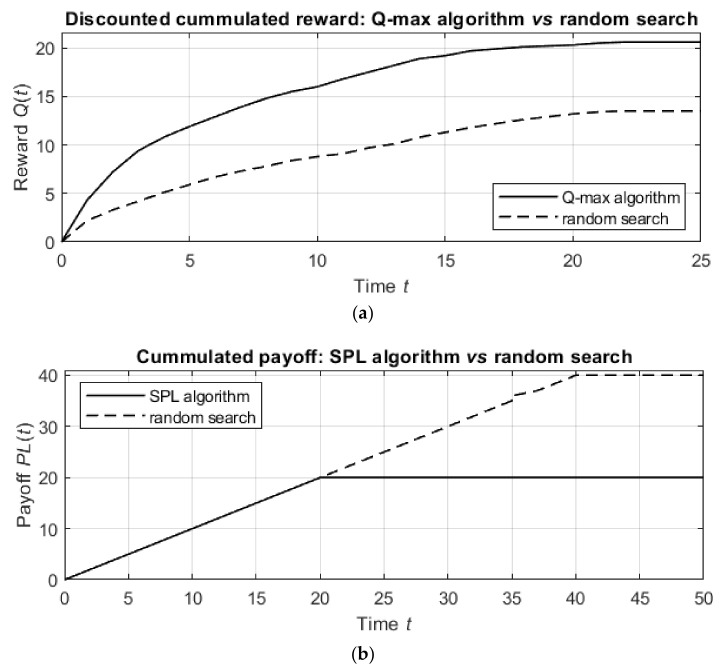
Discounted cumulative reward of detection by the Q-max algorithm (**a**) and cumulative payoff of detection by the SPL algorithm (**b**) compared with the results obtained by the random detection procedure. The solid line in both figures is associated with the suggested algorithms (Q-max and SPL), and the dashed line is associated with the random choice of actions.

**Figure 8 entropy-24-01168-f008:**
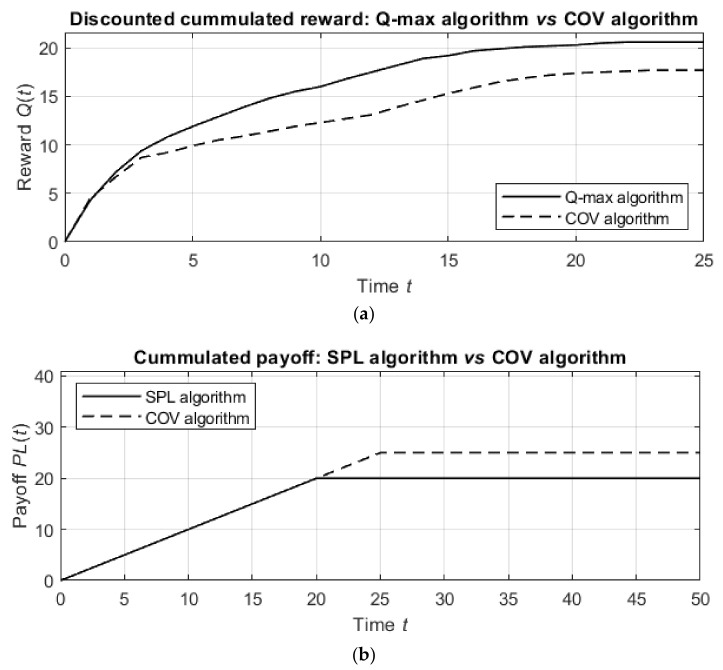
Cumulative reward of detection by the Q-max algorithm for static targets (**a**) and cumulative payoff of detection by the SPL algorithm for static targets (**b**) compared with the results obtained by the COV algorithm.

**Figure 9 entropy-24-01168-f009:**
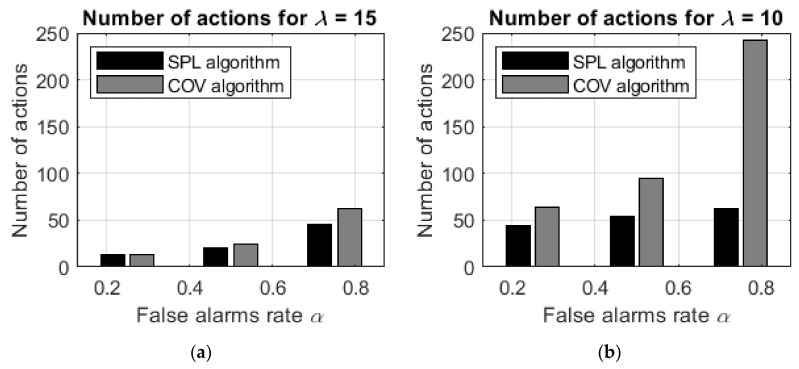
The number of agent actions in detecting two static targets with the SPL/Q-max algorithms (black bars) and the COV algorithm (gray bars): (**a**) λ=15 and (**b**) λ=10.

**Figure 10 entropy-24-01168-f010:**
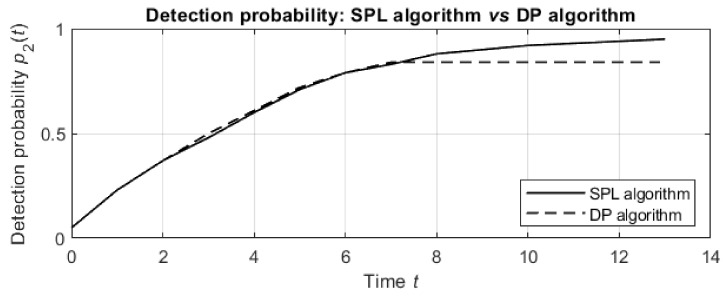
Dependence of the detection probabilities on the number of planned actions for the SPL algorithm (solid line) and DP algorithm (dotted line); the sensor sensitivity is λ=15, the false alarm rate is α=0.25, and the termination time is t=120 min.

**Figure 11 entropy-24-01168-f011:**
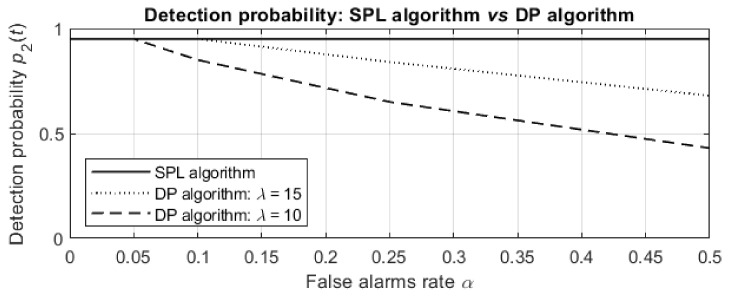
Dependence of the detection probabilities on the false alarm rate α for sensor sensitivities λ=15 (dotted line) and λ=10 (dashed line). The probability 0.95 for the SPL algorithm and all values of α is depicted by the solid line. The termination time is 120 min.

**Table 1 entropy-24-01168-t001:** Number of agent actions and the discounted cumulative information gain in detecting two static targets for the false alarm rate α=0.5.

Detection Algorithm	Number of Actions Up to Detection of the First Target	Number of Actions Up to Detection of the Second Target	Discounted Cumulative Information Gain
Random	25	45	13.4
EIG	17	27	17.1
COV	17	24	17.5
COG	18	29	16.1
Q-max	15	21	20.5
SPL	14	21	20.1

**Table 2 entropy-24-01168-t002:** Number of agent actions and the discounted cumulative information gain in detecting two moving targets for the false alarm rate α=0.5.

Detection Algorithm	Number of Actions Up to Detection of the First Target	Number of Actions Up to Detection of the Second Target	Discounted Cumulative Information Gain
Random	72	105	21.8
EIG	50	65	27.1
COV	49	62	28.7
COG	55	67	26.2
Q-max	32	45	33.2
SPL	31	43	32.1

**Table 3 entropy-24-01168-t003:** The number of agent actions in detecting two static targets for different values of the false alarm rate α and of the sensor sensitivity λ.

Sensor Sensitivity	Algorithm	False Alarm Rate		
		α=0.25	α=0.5	α=0.75
λ=15	COV	14	25	63
	SPL/Q-max (average)	13	20	45
λ=5	COV	64	95	242
	SPL/Q-max (average)	44	54	63

**Table 4 entropy-24-01168-t004:** Number of planned agent actions in detecting two static targets by the SPL algorithm and dynamic programming (DP) algorithm for different values of the false alarm rate α and of the sensor sensitivity λ.

Sensor Sensitivity	Algorithm	Characteristic	False Alarm Rate				
			α=0	α=0.05	α=0.1	α=0.25	α=0.5
λ=15	DP	Run time	0.4 s	1 min	120 min	120 min	120 min
		Number of planned actions	3	5	7	7	7
		Detection probabilities $p_{1}$ $\mathrm{and}\;p_{2}$	1.0 1.0	1.0 0.99	0.99 0.96	0.90 0.84	0.84 0.68
	SPL	Run time	0.4 s	1 min	120 min	120 min	120 min
		Number of planned actions	3	5	7	13	20
		Detection probabilities $p_{1}$ $\mathrm{and}\;p_{2}$	1.0 1.0	1.0 0.99	0.99 0.96	0.99 0.95	0.99 0.95
λ=10	DP	Run time	1 min	120 min	120 min	120 min	120 min
		Number of planned actions	5	7	7	7	7
		Detection probabilities $p_{1}$ $\mathrm{and}\;p_{2}$	1.0 1.0	0.96 0.95	0.90 0.85	0.85 0.65	0.71 0.43
	SPL	Run time	1 min	120 min	120 min	120 min	120 min
		Number of planned actions	5	7	15	21	32
		Detection probabilities $p_{1}$ $\mathrm{and}\;p_{2}$	1.0 1.0	0.96 0.95	0.97 0.95	0.98 0.95	0.99 0.95

**Table 5 entropy-24-01168-t005:** Run times and temporal difference errors with respect to the size of the data set.

Domain Size $n_{x}\times n_{y}$	Number of Nonzero Weights in the Neural Network	Size of the Data Set	Run Time for One Epoch [Minutes]	Mean Squared Error *
10×10	42,009	5000	4	0.13
		10,000	8	0.12
20×20	648,009	5000	7	0.15
		10,000	14	0.13
40×40	10,272,009	5000	10	0.18
		10,000	20	0.15

* The error was calculated over the temporal difference errors at the validation stage at epoch t=30.

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
