# Peer review of "Detection of Static and Mobile Targets by an Autonomous Agent with Deep Q-Learning Abilities"

_entropy, 2022, doi:10.3390/e24081168_

Round 1
Reviewer 1 Report
Good paper presentation.
Formula are very good described. I suggest to expand bibliography in order give more scientific sound to your paper.
Reviewer 2 Report
The topic is relevant and may be of interest to a broad range of the journal's readers. However, this reviewer has some major concerns about the paper.
Major Strengths: The major strengths of the research are:
The proposed approach addresses the problem of detecting multiple static and mobile targets by an autonomous mobile agent acting under uncertainty. Authors assume that an agent is able to detect targets at different distances by also keeping track of detection errors.
Major Weaknesses: The major weaknesses of the research are:
- The novel contributions of the proposal are not clear. I suggest clearly describing them in the Introduction section.
- The procedures included in the paper should be represented as pseudocodes and not as a set of sentences with a bulleted list.
- Tables and Figures should be inserted in the margins of the pages
- My main concern is the innovativeness of the proposal with respect to the state of the art. I suggest discussing it, by also introducing a new experimental evaluation to compare the proposed approach with another approach in the leterature.
- The flow in figure 3 is not clear.
- A related work section is required in order to highlight the novelty of the proposal.
- A suggestion for future work: it would be interesting to investigate how data preparation approaches affect the results of the proposed algorithms. For instance, the application of feature engineering approaches, such as https://doi.org/10.1016/j.eswa.2022.117957 or https://doi.org/10.1007/s11227-021-04250-0, or the profiling techniques for extracting metadata from data and repair it (https://doi.org/10.1145/3487664.3487719), could affect the overall quality of the results.
Grammar and Readability:
The paper is well written and clear. I don't found any typo.
Specific Comments: My specific comments concerning this manuscript are:
I think that it is necessary to improve the overall quality of the paper before proceeding with the publication by following the considerations reported in the review.
Reviewer 3 Report
This paper proposed a novel Q-learning-based AI method that can address an important issue in autonomous driving ---- multitarget detection. The paper is interesting. But, Q learning isn't a new idea. Therefore, the authors should still consider highlighting their own special contributions. The overall quality is good. But, the authors may consider the following comments to further improve the paper:
See (2), why you use the very wordy way to describe '?????? ??????? ?? ??'. You can totally use a symbol like ci^l to denote it. The authors used -> to represent actions. It indicates that the authors have the capability to use simple ways to describe things clearly.
Line 103, what's ~?
I think (5) can also be improved.
Why did you write || not | in (14)?
In line 203 and the following pseudo-code, the authors mentioned asymmetric sigmoid activation function. But the equation looks like a very normal and typical sigmoid function. Why is that?
What does 'playoff' means? You mentioned 'payoff represents the steps of the agent; ' but that's way too confusing.
Before you talk about what software and what hardware you used for the simulation, please specify what tasks you are doing in the simulation. For example, what targets you're detecting. This is a very important flaw of this paper.
You may need to improve the references. At least read Sutton, Richard S., and Andrew G. Barto. Reinforcement learning: An introduction. MIT press, 2018.
All in all, this paper proposed good architecture and put a lot of effort into it. I have no objection to the publication. But, please consider the comments given above.
